# CERTIFIED DEFENSES FOR ADVERSARIAL PATCHES*

**Ping-yeh Chiang**[†]**, Renkun Ni**[†]**, Ahmed Abdelkader, Chen Zhu**
University of Maryland, College Park
{pchiang,rn9zm,akader,chenzhu}@cs.umd.edu

**Christoph Studer**
Cornell University
studer@cornell.edu

**Tom Goldstein**
University of Maryland, College Park
tomg@cs.umd.edu

## ABSTRACT

Adversarial patch attacks are among one of the most practical threat models against real-world computer vision systems. This paper studies certified and empirical defenses against patch attacks. We begin with a set of experiments showing that most existing defenses, which work by pre-processing input images to mitigate adversarial patches, are easily broken by simple white-box adversaries. Motivated by this finding, we propose the first certified defense against patch attacks, and propose faster methods for its training. Furthermore, we experiment with different patch shapes for testing, obtaining surprisingly good robustness transfer across shapes, and present preliminary results on certified defense against sparse attacks. Our complete implementation can be found on: https://github.com/Ping-C/certifiedpatchdefense.

## 1 INTRODUCTION

Despite the great success of neural networks for vision problems, they are easily fooled by adversarial attacks in which the input to a machine learning model is modified with the goal of manipulating its output. Research in this area is largely focused on norm-bounded attack (Madry et al., 2017; Tramèr & Boneh, 2019; Shafahi et al., 2019), where the adversary is allowed to perturb *all* pixels in an image provided that the $\ell_p$-norm of the perturbation is within prescribed bounds. Other adversarial models were also proposed, such as functional (Laidlaw & Feizi, 2019), rotation/translation (Engstrom et al., 2017), and Wasserstein (Wong et al., 2019), all of which allow modification to all pixels.

Whole-image perturbations are unrealistic for modeling "physical-world" attacks, in which a real-world object is modified to evade detection. A physical adversary usually modifies an object using stickers or paint. Because this object may only occupy a small portion of an image, the adversary can only manipulate a limited number of pixels. As such, the more practical *patch attack* model was proposed (Brown et al., 2017). In a patch attack, the adversary may only change the pixels in a confined region, but is otherwise free to choose the values yielding the strongest attack. The threat to real-world computer vision systems is well-demonstrated in recent literature where carefully crafted patches can fool a classifier with high reliability (Brown et al., 2017; Karmon et al., 2018), make objects invisible to an object detector (Wu et al., 2019; Lee & Kolter, 2019), or fool a face recognition system (Sharif et al., 2017). In light of such effective physical-world patch attacks, very few defenses are known to date.

In this paper, we study principled defenses against patch attacks. We begin by looking at existing defenses in the literature that claim to be effective against patch attacks, including Local Gradient Smoothing (LGS) (Naseer et al., 2019) and Digital Watermarking (DW) (Hayes, 2018). Similar to what has been observed for whole-image attacks by (Athalye et al., 2018), we show that these patch defenses are easily broken by stronger adversaries. Concretely, we demonstrate successful white-box attacks, where the adversary designs an attack against a known model, including any pre-processing

---

*This work was supported by the DARPA GARD, DARPA QED4RML programs, and National Science Foundation DMS division.

[†]equal contribution

steps. To cope with such potentially stronger adversaries, we train a robust model that produces a lower-bound on adversarial accuracy. In particular, we propose the first *certifiable* defense against patch attacks by extending interval bound propagation (IBP) defenses (Gowal et al., 2018; Mirman et al., 2018). We also propose modifications to IBP training to make it faster in the patch setting. Furthermore, we study the generalization of certified patch defenses to patches of different shapes, and observe that robustness transfers well across different patch types. We also present preliminary results on certified defense against the stronger sparse attack model, where a fixed number of possibly non-adjacent pixels can be arbitrarily modified (Modas et al., 2019).

## 2 PROBLEM SETUP

We consider a white-box adversary that is allowed to choose the location of the patch (chosen from a set $\mathbb{L}$ of possible locations) and can modify pixels within the particular patch (chosen from the set $\mathbb{P}$) similar to (Karmon et al., 2018). An attack is successful if the adversary changes the classification of the network to a wrong label. In this paper, we are primarily interested in the patch attack robust accuracy (adversarial accuracy for short) as defined by

$$\mathbb{E}_{x \sim X} \min_{p \in \mathbb{P}, l \in \mathbb{L}} \mathcal{X}[f(A(x, p, l); \theta) = y], \tag{1}$$

where the operator $A$ places the adversarial patch p on a given image x at location $l$, f is a neural network with parameter $\theta$, $X$ is a distribution of images, and $\mathcal{X}$ is a characteristic function that takes value 1 if its argument is true, and 0 otherwise.

In this model, the strength of the adversary can vary depending on the set of possible patches allowed, and the type of perturbation allowed within the patch. In what follows, we assume the standard setup in which the adversary is allowed any perturbation that maintains pixel intensities in the range $[0, 1]$. Unless otherwise noted, we also assume the patch is restricted to a square of prescribed size. We consider two different options for the set $\mathbb{L}$ of possible patch locations. First, we consider a weak adversary that can only place patches at the corner of an image. We find that even this weak model is enough to break existing patch defenses. Then, we consider a stronger adversary with no restrictions on patch location, and use this model to evaluate our proposed defenses. Note that an adversary, when restricted to modify only a square patch at location $l$ in the image, has the freedom to modify any non-square subset of these pixels. In other words, a certified defense against square patch attacks also provably subverts any non-square patch attack that fits inside a small enough square.

In general, calculating the adversarial accuracy (1) is intractable due to non-convexity. Common approaches try to approximate it by solving the inner minimization using a gradient-based method. However, in Section 3, we show that depending on how the minimization is solved, the upper bound could be very loose: a model may appear to be very robust, but fail when faced with a stronger attack. To side-step the arms race between attacks and defenses, in Section 4, we extend the work of (Gowal et al., 2018) and (Mirman et al., 2018) to train a network that produces a lower bound on adversarial accuracy. We will refer to approximations of the upper bound as *empirical adversarial accuracy* and the lower bound as *certified accuracy*.

## 3 VULNERABILITY OF EXISTING DEFENSES

We start by examining existing defense strategies that claim to be effective against patch attacks. Similar to what has been observed for whole-image attacks by Athalye et al. (2018), we show that these patch defenses can easily be broken by white-box attacks, where the adversary optimizes against a given model including any pre-processing steps.

### 3.1 EXISTING DEFENSES

Under our threat model, two defenses have been proposed that each use input transformations to detect and remove adversarial patches.

The first defense is based on the observation that the gradient of the loss with respect to the input image often exhibits large values near the perturbed pixels. In (Hayes, 2018), the proposed digital watermarking (DW) approach exploits this behavior to detect unusually dense regions of large

Table 1: Empirical adversarial accuracy of ImageNet classifiers defended with Local Gradient Smoothing and Digital Watermarking. We consider two types of adversaries, one that takes the defense into account during backpropagation and one that does not

| | | Patch Size | | |
|---|---|---|---|---|
| Attack | Defense | $42 \times 42$ | $52 \times 52$ | $60 \times 60$ |
| IFGSM | LGS | 78% | 75% | 71% |
| IFGSM + LGS | LGS | 14% | 5% | 3% |
| IFGSM | DW | 56% | 49% | 45% |
| IFGSM + DW | DW | 13% | 8% | 5% |

gradient entries using saliency maps, before masking them out in the image. Despite a $12\%$ drop in accuracy on clean (non-adversarial) images, this defense method supposedly achieves an empirical adversarial accuracy of $63\%$ for non-targeted patch attacks of size $42 \times 42$ ($2\%$ of the image pixels), using 400 randomly picked images from ImageNet (Deng et al., 2009) on VGG19 (Simonyan & Zisserman, 2014).

The second defense, Local Gradient Smoothing (LGS) by Naseer et al. (2019) is based on the empirical observation that pixel values tend to change sharply within these adversarial patches. In other words, the image gradients tend to be large within these adversarial patches. Note that the image gradient here differs from the gradient in Hayes (2018), the former is with respect the changes of adjacent pixel values and the later is with respect to the classification loss. Naseer et al. (2019) propose suppressing this adversarial noise by multiplying each pixel with one minus its image gradient as in (2). To make their methods more effective, Naseer et al. (2019) also pre-process the image gradient with a normalization and a thresholding step.

$$\hat{x} = x \odot (1 - \lambda g(x)). \tag{2}$$

The $\lambda$ here is a smoothing hyper-parameter. Naseer et al. (2019) claim the best adversarial accuracy on ImageNet with respect to patch attacks among all of the defenses we studied. They also claim that their defense is resilient to Backward Pass Differential Approximation (BPDA) from Athalye et al. (2018), one of the most effective methods to attack models that include a non-differentiable operator as a pre-processing step.

### 3.2 Breaking Existing Defenses

Using a similar setup as in (Hayes, 2018; Naseer et al., 2019), we are able to mostly replicate the reported empirical adversarial accuracy for Iterative Fast Gradient Sign Method (IFGSM), a common gradient based attack, but we show that when the pre-processing step is taken into account, the empirical adversarial accuracy on ImageNet quickly drops from $\sim 70\%(\sim 50\%)$ for LGS(DW) to levels around $\sim 10\%$ as shown in Table 1.

Specifically, we break DW (Hayes, 2018) by applying BPDA, in which the non-differentiable operator is approximated with an identity mapping during the backward pass. We break LGS (Naseer et al., 2019) by directly incorporating the smoothing step during backpropagation. Even though the windowing and thresholding steps are non-differentiable, the smoothing operator provides enough gradient information for the attack to be effective.

To make sure that our evaluation is fair, we used the exact same models as Hayes (2018) (VGG19) and Szegedy et al. (2016) (Inception V3). We also consider a weaker set of attackers that can only attack the corners, the same as their setting. Further, we ensure that we were able to replicate their reported result under similar setting.

### 4 Certified Defenses

Given the ease with which these supposedly strong defenses are broken, it is natural to seek methods that can rigorously guarantee robustness of a given model to patch attacks. With such *certifiable* guarantees in hand, we need not worry about an adversary with a stronger optimizer, or a more clever algorithm for choosing patch locations.

## 4.1 Background on certified defenses

Certified defenses have been intensely studied with respect to norm-bounded attacks (Cohen et al., 2019; Wong & Kolter, 2017; Gowal et al., 2018; Mirman et al., 2018; Zhang et al., 2019b). In all of these methods, in addition to the prediction model, there is also a verifier. Given a model and an input, the verifier outputs a certificate if it is guaranteed that the image can not be adversarially perturbed. This is done by checking whether there exists any nearby image (within a prescribed $\ell_p$ distance) with a different label than the image being classified. While theoretical bounds exist on the size of this distance that hold for any classifier (Shafahi et al., 2018), exactly computing bounds for a specific classifier and test image is hard. Alternatively, the verifier may output a lower bound on the distance to the nearest image of a different label. This latter distance is referred to as the *certifiable radius*. Most of these verifiers provide a rather loose bound on the certifiable radius. However, if the verifier is differentiable, then the network can be trained with a loss that promotes tightness of this bound. We use the term *certificate training* to refer to the process of training with a loss that promotes strong certificates. Interval bound propagation (IBP) (Mirman et al., 2018; Gowal et al., 2018) is a very simple verifier that uses layer-wise interval arithmetic to produce a certificate. Even though the IBP certificate is generally loose, after certificate training, it yields state-of-the-art certifiably-robust models for $l_\infty$-norm bounded attacks (Gowal et al., 2018; Zhang et al., 2019b). In this paper, we extend IBP to train certifiably-robust networks resilient to patch attacks. We first introduce some notation and basic algorithms for IBP training.

**Notation** We represent a neural network with a series of transformations $h^{(k)}$ for each of its $k$ layers. We use $z^{(k)} \in \mathbb{R}^{n_k}$ to denote the output of layer $k$, where $n_k$ is the number of units in the $k^{th}$ layer and $z^{(0)}$ stands for the input. Specifically, the network computes

$$z^{(k)} = h^{(k-1)}(z^{(k-1)}) \qquad \forall k = 1, \dots, K.$$

**Certification Problem** To produce a certificate for an input $x_0$, we want to verify that the following condition is true with respect to all possible labels $y$:

$$(e_{y_{true}} - e_y)^T z^{(K)} = \mathbf{m}_y \geq 0 \qquad \forall z^{(0)} \in \mathbb{B}(x_0) \qquad \forall y. \tag{3}$$

Here, $e_i$ is the $i^{th}$ basis vector, and $\mathbf{m}_y$ is called the margin following Wong & Kolter (2017). Note that $\mathbf{m}_{y_{true}}$ is always equal to 0. The vector $\mathbf{m}$ contains all margins corresponding to all labels. $\mathbb{B}(x_0)$ is the constraint set over which the adversarial input image may range. In a conventional setting, this is an $\ell_\infty$ ball around $x_0$. In the case of patch attack, the constraint set contains all images formed by applying a patch to $x_0$;

$$\mathbb{B}(x_0) = \{A(x_0, p, l) | p \in \mathbb{P} \text{ and } l \in \mathbb{L}\}. \tag{4}$$

**The Basics of Interval Bound Propagation (IBP)** We now describe how to produce certificates using interval bound propagation as in (Gowal et al., 2018). Suppose that for each component in $z^{(k-1)}$ we have an interval containing all the values which this component reaches as $z^{(0)}$ ranges over the ball $\mathbb{B}(x_0)$. If $z^{(k)} = h^{(k)}(z^{(k-1)})$ is a linear (or convolutional) layer of the form $z^{(k)} = W^{(k)} z^{(k-1)} + b^{(k)}$, then we can get an *outer approximation* of the reachable interval range of activations by the next layer $z^{(k)}$ using the formulas below

$$\overline{z}^{(k)} = W^{(k)} \frac{\overline{z}^{(k-1)} + \underline{z}^{(k-1)}}{2} + |W^{(k)}| \frac{\overline{z}^{(k-1)} - \underline{z}^{(k-1)}}{2} + b^{(k)}, \tag{5}$$

$$\underline{z}^{(k)} = W^{(k)} \frac{\overline{z}^{(k-1)} + \underline{z}^{(k-1)}}{2} - |W^{(k)}| \frac{\overline{z}^{(k-1)} - \underline{z}^{(k-1)}}{2} + b^{(k)}. \tag{6}$$

Here $\overline{z}^{(k-1)}$ denotes the upper bound of each interval, $\underline{z}^{(k-1)}$ the lower bound, and $|W^{(k)}|$ the element-wise absolute value. Alternatively, if $h^{(k)}(z^{(k-1)})$ is an element-wise monotonic activation (e.g., a ReLU), then we can calculate the *outer approximation* of the reachable interval range of the next layer using the formulas below.

$$\overline{z}^{(k)} = h^{(k)}(\overline{z}^{(k-1)}) \tag{7}$$

$$\underline{z}^{(k)} = h^{(k)}(\underline{z}^{(k-1)}). \tag{8}$$

When the feasible set $\mathbb{B}(x_0)$ represents a simple $\ell_\infty$ attack, the range of possible $z^{(0)}$ values is trivially characterized by an interval bound $\overline{z}^{(0)}$ and $\underline{z}^{(0)}$. Then, by iteratively applying the above rules, we can propagate intervals through the network and eventually get $\overline{z}^{(K)}$ and $\underline{z}^{(K)}$. A certificate can then be given if we can show that (3) is always true for outputs in the range $\overline{z}^{(K)}$ and $\underline{z}^{(K)}$ with respect to all possible labels. More specifically, we can check that the following holds for all $y$

$$\underline{\mathbf{m}}_y = e_{y_{true}}^T \underline{z}^{(K)} - e_y^T \overline{z}^{(K)} = \underline{z}_{y_{true}}^{(K)} - \overline{z}_y^{(K)} \geq 0 \quad \forall y. \tag{9}$$

**Training for Interval Bound Propagation** To train a network to produce accurate interval bounds, we simply replace standard logits with the $-\underline{\mathbf{m}}$ vector in (3). Note that all elements of $\underline{\mathbf{m}}$ need to be larger than zero to satisfy the conditions in (3), and $\underline{\mathbf{m}}_{y_{true}}$ is always equal to zero. Put simply, we would like $\underline{\mathbf{m}}_{y_{true}} = 0$ to be the least of all margins. We can promote this condition by training with the loss function

$$\text{Certificate Loss} = \text{Cross Entropy Loss}(-\underline{\mathbf{m}}, y). \tag{10}$$

Unlike regular neural network training, stochastic gradient descent for minimizing equation 10 is unstable, and a range of tricks are necessary to stabilize IBP training (Gowal et al., 2018). The first trick is merging the last linear weight matrix with $(e_y - e_{y_{true}})$ before calculating $-\underline{\mathbf{m}}_y$. This allows a tighter characterization of the interval bound that noticeably improves results. The second trick uses an "epsilon schedule" in which training begins with a perturbation radius of zero, and this radius is slowly increased over time until a sentinel value is reached. Finally, a mixed loss function containing both a standard natural loss and an IBP loss is used.

In all of our experiments, we use the merging technique and the epsilon schedule, but we do not use a mixed loss function containing a natural loss as it does not increase our certificate performance.

## 4.2 CERTIFYING AGAINST PATCH ATTACKS

We can now describe the extension of IBP to patches. If we specify the patch location, one can represent the feasible set of images with a simple interval bound: for pixels within the patch, the upper and lower bound is equal to 1 and 0. For pixels outside of the patch, the upper and lower bounds are both equal to the original pixel value. By passing this bound through the network, we would be able to get $\underline{\mathbf{m}}^{\text{single location}}$ and verify that they satisfy the conditions in (3).

However, we have to consider not just a single location, but all possible locations $\mathbb{L}$ to give a certificate. To adapt the bound to all possible location, we pass each of the possible patches through the network, and take the worst case margin. More specifically,

$$\underline{\mathbf{m}}^{\text{es}}(\mathbb{L})_y = \min_{l \in \mathbb{L}} \underline{\mathbf{m}}^{\text{single patch}}(l)_y \quad \forall y. \tag{11}$$

Similar to regular IBP training, we simply use $\underline{\mathbf{m}}^{\text{es}}(\mathbb{L})$ to calculate the cross entropy loss for training and backpropagation,

$$\text{Certificate Loss} = \text{Cross Entropy Loss}(-\underline{\mathbf{m}}^{\text{es}}(\mathbb{L}), y). \tag{12}$$

Unfortunately, the cost of producing this naïve certificate increases quadratically with image size. Consider that a CIFAR-10 image is of size $32 \times 32$, requiring over a thousand interval bounds, one for each possible patch location. To alleviate this problem, we propose two certificate training methods: *Random Patch* and *Guided Patch*, so that the number of forward passes does not scale with the dimension of the inputs.

**Random Patch Certificate Training** In this method, we simply select a random set of patches out of the possible patches and pass them forward. A level of robustness is achieved even though a very small number of random patches are selected compared to the total number of possible patches

$$\mathbf{m}^{\text{random patches}}(\mathbb{L})_y = \underline{\mathbf{m}}^{\text{es}}(S)_y \tag{13}$$

where $S$ is a random subset of $\mathbb{L}$. Similarly, the random patch certificate loss is calculated as below.

$$\text{Random Patch Certificate Loss} = \text{Cross Entropy Loss}(-\mathbf{m}^{\text{random patches}}(\mathbb{L}), y) \tag{14}$$

**Guided Patch Certificate Training** In this method, we propose using a U-net (Ronneberger et al., 2015) to predict $\underline{\mathbf{m}}^{\text{single patch}}$, and then randomly select a couple of locations based on the predicted $\underline{\mathbf{m}}^{\text{single patch}}$ so that fewer patches need to be passed forward.

Note that very few patches contribute to the worst case bound $\underline{\mathbf{m}}^{\text{es}}$ in (11). In fact, the number of patches that yield the worst case margins will be no more than the number of labels. If we know the worst-case patches beforehand, then we can simply select the few worst-case patches during training.

We propose to use U-net as the number of locations and margins is very large. For a square patch of size $n \times n$ and an image of size $m \times m$, the total number of possible locations is $(m - n + 1)^2$, and for each location the number of margins is equal to the number of possible labels.

$$\underline{\mathbf{m}}^{\text{pred}} = \text{U-net(image)} \tag{15}$$

$$\dim(\underline{\mathbf{m}}^{\text{pred}}) = (m - n + 1, m - n + 1, \# \text{ of labels}). \tag{16}$$

Given the U-net prediction of $\underline{\mathbf{m}}^{\text{pred}}$, we then randomly select a single patch for each label based on the softmax of the predicted $\underline{\mathbf{m}}^{\text{pred}}$. The number of selected patches is equal to the number of labels. After these patches are passed forward, the U-net is then updated with a mean-squared-error loss between the predicted margins $\underline{\mathbf{m}}^{\text{pred}}$ and the actual margins $\underline{\mathbf{m}}^{\text{actual}}$. Note that only a few patches are selected at a time, so that the mean-squared-error only passes through the selected patches.

$$\text{U-net Loss} = \text{MSE}(\underline{\mathbf{m}}^{\text{pred}}, \underline{\mathbf{m}}^{\text{actual}}). \tag{17}$$

The network is trained with the following loss:

$$\text{Guided Patch Certificate Loss} = \text{Cross Entropy Loss}(-\underline{\mathbf{m}}^{\text{guided patches}}(\mathbb{L}), y). \tag{18}$$

**Certification Process** In all our experiments, we check that equation (3) is satisfied by iterating over all possible patches and forward-passing the interval bounds generated for each patch; this overhead is tolerable at evaluation time.

### 4.3 CERTIFYING AGAINST SPARSE ATTACKS

IBP can also be adapted to defend against sparse attack where the attacker is allowed to modify a fixed number ($k$) of pixels that may not be adjacent to each other (Modas et al., 2019). The only modification is that we have to change the bound calculated from the first layer to

$$\overline{z}_i^{(1)} = W_{i,:}^{(1)} z^{(0)} + |W_{i,:}^{(1)}|_{top_k} \qquad \underline{z}_i^{(1)} = W_{i,:}^{(1)} z^{(0)} - |W_{i,:}^{(1)}|_{top_k} \qquad \forall i \tag{19}$$

and apply equation (5) and (6) for the subsequent layers. Here, $(.)_{top_k}$ is the sum of the largest $k$ elements in the vector.

## 5 EXPERIMENTS

In this section, we compare our certified defenses with exiting algorithms on two datasets and three model architectures of varying complexity. We consider a strong attack setting in which adversarial patches can appear anywhere in the image. Different training strategies for the certified defense are also compared, which shows a trade-off between performance and training efficiency. Furthermore, we evaluate the transferability of a model trained using square patches to other adversarial shapes, including shapes that do not fit in any certified square. The training and architectural details can be found in Appendix A.1. We also present preliminary results on certified defense against sparse attacks.

### 5.1 COMPARISON AGAINST EXISTING DEFENSES

In this section, we study the effectiveness of our proposed IBP certified models against an adversary that is allowed to place patches anywhere in the image, even on top of the salient object. If the patch is sufficiently small, and does not cover a large portion of the salient object, then the model should still classify correctly, and defense against the perturbation should be possible.

In the best case, our IBP certified model is able to achieve 91.6% certified (Table 2) with respect to a 2×2 patch ($\sim$ .5% of image pixels) adversary on MNIST. For more challenging cases, such as a 5

$\times 5$ ( $\sim 2.5\%$ of image pixels) patch adversary on CIFAR-10, the certified adversarial accuracy is only 24.9% (Table 2). Even though these existing defenses appear to achieve better or comparable adversarial accuracy as our IBP certified model when faced with a weak adversary, when faced with a stronger adversary their adversarial accuracy dropped to levels below our certified accuracy for all cases that we analyzed.

When evaluating existing defenses, we only report cases where non-trivial adversarial accuracy is achieved against a weaker adversary. We do not explore cases where LGS and DW perform so poorly that no meaningful comparison can be done. LGS and DW are highly dependent on hyperparameters to work effectively against naive attacks, and yet neither Naseer et al. (2019) nor Hayes (2018) proposed a way to learn these hyperparameters. By trial and error, we were able to increase the adversarial accuracy against a weaker adversary for some settings, but not all. In addition, we also notice a peculiar feature of DW: when we increase the adversarial accuracy, the clean accuracy degrades, sometimes so much that it is even lower than the empirical adversarial accuracy. This happens because DW always removes a patch from the prediction. When an adversarial patch is detected, it is likely to be removed, enabling correct prediction. On the other hand, when there are no adversarial patches, DW removes actual salient information, resulting in lower clean accuracy.

Here we did not compare our results with adversarial training, because even though it produces some of the most adversarially robust models, it does not offer any guarantees on the empirical robust accuracy, and could still be decreased further with stronger attacks. For example, Wang et al. (2019) proposed a stronger attack that could find 47% more adversarial examples compared to gradient based method. Further, adversarial training on all possible patches would be even more expensive compared to certificate training, and is slightly beyond our computational budget.

Compared to state-of-the-art certified models for CIFAR with $L_\infty$-perturbation, where Zhang et al. (2019a) proposed a deterministic algorithm that achieves clean accuracy of $34.0\%$, our clean accuracy for our most robust CIFAR $5 \times 5$ model is $47.8\%$ when using a large model (Table 2).

Table 2: Comparison of our IBP certified patch defense against existing defenses. Empirical adversarial accuracy is calculated for 400 random images in both datasets. All results are averaged over three different models.

| Dataset | Patch Size | Adversary | Defense | Clean Accuracy | Empirical Adversarial Accuracy | Certified Accuracy |
|---|---|---|---|---|---|---|
| MNIST | $2 \times 2$ | IFGSM | None | 98.4% | 80.1% | - |
| | $2 \times 2$ | IFGSM | LGS | 97.4% | 90.0% | - |
| | $2 \times 2$ | IFGSM + LGS | LGS | 97.4% | 60.7% | - |
| | $2 \times 2$ | IFGSM | IBP | 98.5% | 93.9% | 91.6% |
| | $5 \times 5$ | IFGSM | None | 98.5% | 3.3% | - |
| | $5 \times 5$ | IFGSM | IBP | 92.9% | 66.1% | 62.0% |
| CIFAR | $2 \times 2$ | IFGSM | None | 66.3% | 25.4% | - |
| | $2 \times 2$ | IFGSM | LGS | 64.9% | 31.3% | - |
| | $2 \times 2$ | IFGSM + LGS | LGS | 64.9% | 24.2% | - |
| | $2 \times 2$ | IFGSM | DW | 47.1% | 43.3% | - |
| | $2 \times 2$ | IFGSM + DW | DW | 47.1% | 20.2% | - |
| | $2 \times 2$ | IFGSM | IBP | 48.6% | 45.2% | 41.6% |
| | $5 \times 5$ | IFGSM | None | 66.5% | 0.4% | - |
| | $5 \times 5$ | IFGSM | LGS | 51.2% | 22.11% | - |
| | $5 \times 5$ | IFGSM + LGS | LGS | 51.2% | 0.5% | - |
| | $5 \times 5$ | IFGSM | DW | 45.3% | 59.3% | - |
| | $5 \times 5$ | IFGSM + DW | DW | 45.3% | 15.6% | - |
| | $5 \times 5$ | IFGSM | IBP | 33.9% | 29.1% | 24.9% |

## 5.2 COMPARISON OF TRAINING STRATEGIES

We find that given a fixed architecture all-patch certificate training achieves the best certified accuracy. However, given a fixed computational budget, random and guided training significantly outperform all-patch training. Finally, guided-patch certificate training consistently outperforms random-patch

certificate training by a slim margin, indicating that the U-net is learning how to predict the minimum margin $\underline{\mathbf{m}}$.

In Table 3, we see that given a fixed architecture all-patch certificate training significantly outperforms both random-patch certificate training and guided-patch certificate training in terms of certified accuracy, outperforming the second best certified defenses in each task by 2.6% (MNIST, $2 \times 2$), 7.3% (MNIST, $5 \times 5$), 3.9% (CIFAR-10, $2 \times 2$), and 3.4% (CIFAR-10, $5 \times 5$). However, all-patch certificate training is very expensive, taking on average 4 to 15 times longer than guided-patch certificate training and over 30 to 70 times longer than random-patch certificate training.

On the other hand, given a limited computational budget, random-patch and guided-patch training significantly outperforms all-patch training. Due to the efficiency of random-patch and guided-patch training, they scale much better to large architectures. By switching to a large architecture (5 layer wide convolutional network), we are able to boost the certified accuracy by over 10% compared to the best performing all-patch small model (Table 2). Note that we are unable to all-patch train the same architecture as it will take almost 15 days to complete, and is out of our computational budget.

Guided-patch certificate training is slightly more expensive compared to random patch, due to overhead from the U-net architecture. However, given the 10 patches picked, guided-patch certificate training consistently outperforms random-patch certificate training, indicating that the U-net is learning how to predict the minimum margin $\underline{\mathbf{m}}$.

Table 3: Trade-off between certified accuracy and training time for different strategies. The numbers next to training strategies indicate the number of patches used for estimating the lower bound during training. Most training times are measured on a single 2080Ti GPU, with the exception of all-patch training which is run on four 2080Ti GPUs. For that specific case, the training time is multiplied by 4 for fair comparison. See Appendix A.6 for more detailed statistics. *indicates the performance of the best performing large model trained with either random or guided patch. Detailed performance of the large models can be found in Appendix A.5

| Dataset | Training Strategy | $2 \times 2$ | | | $5 \times 5$ | | |
|---|---|---|---|---|---|---|---|
| | | Clean Accuracy | Certified Accuracy | Training Time(h) | Clean Accuracy | Certified Accuracy | Training Time(h) |
| MNIST | All Patch | 98.5% | 91.5% | 9.3 | 92.0% | 60.4% | 8.4 |
| | Random(1) | 98.5% | 82.9% | 0.2 | 96.9% | 24.1% | 0.4 |
| | Random(5) | 98.6% | 86.6% | 0.3 | 95.8% | 42.1% | 0.3 |
| | Random(10) | 98.6% | 87.7% | 0.3 | 95.6% | 49.6% | 0.3 |
| | Guided(10) | 98.6% | 88.9% | 2.2 | 95.0% | 53.1% | 2.6 |
| CIFAR | All Patch | 50.9% | 39.9% | 56.4 | 33.5% | 22.0% | 45.8 |
| | Random(1) | 53.6% | 21.6% | 0.6 | 43.6% | 6.1% | 0.6 |
| | Random(5) | 52.9% | 32.3% | 0.7 | 39.0% | 14.6% | 0.7 |
| | Random(10) | 51.9% | 35.6% | 0.8 | 38.8% | 18.6% | 0.8 |
| | Guided(10) | 52.4% | 36.0% | 3.7 | 37.9% | 18.8% | 3.7 |
| | Large Model* | **65.8%** | **51.9%** | 22.4 | **47.8%** | **30.3%** | 15.4 |

## 5.3 EFFECTIVENESS AGAINST SPARSE ATTACK

The IBP based method can also be used to defend against sparse attack, see Section 4.3. Its performance is reasonable compared to patch defense (e.g. 91.5% certified accuracy for $2 \times 2$ patch vs 90.8% for k=4), even though the sparse attack model is much stronger. For convolutional networks, we increase the size of the first convolutional layer (i.e. from $3 \times 3$ to $11 \times 11$) so the interval bounds calculated are tighter. However, despite the change, fully-connected network still performs much better. For example, the certified accuracy drops from 25.6% to 13.8% when we switch from fully-connected to convolutional network for CIFAR10 and drops from 90.8% to 75.9% for MNIST respectively. Detailed results are shown in the Appendix A.4 Table 7.

Table 4 compares our approach with the state-of-the-art certified sparse defense (Random Ablation) Levine & Feizi (2019). We use their best model with the largest medium radii to certify against various levels of sparsity. As shown in the table, our method achieves higher certified accuracy on the MNIST dataset over all the sparse radii, but lower on CIFAR-10. It is worth noting that we are

using a much smaller and simpler model (a fully-connected network) compared to Random Ablation, which uses ResNet-50.

Table 4: Certified accuracy for sparse defenses with IBP and Random Ablation.

| Dataset | Sparsity ($k$) | Model | Clean Accuracy | Certified Accuracy |
|---|---|---|---|---|
| MNIST | 1 | IBP-sparse | 98.4% | 96.0% |
| | 4 | IBP-sparse | 97.8% | 90.8% |
| | 10 | IBP-sparse | 95.2% | 86.8% |
| | 1 | Random Ablation | 96.7% | 90.3% |
| | 4 | Random Ablation | 96.7% | 79.1% |
| | 10 | Random Ablation | 96.7% | 29.2% |
| CIFAR | 1 | IBP-sparse | 48.4% | 40.0% |
| | 4 | IBP-sparse | 42.2% | 31.2% |
| | 10 | IBP-sparse | 37.0% | 25.6% |
| | 1 | Random Ablation | 78.3% | 68.6% |
| | 4 | Random Ablation | 78.3% | 61.3% |
| | 10 | Random Ablation | 78.3% | 45.0% |

## 5.4 TRANSFERABILITY TO PATCHES OF DIFFERENT SHAPES

Since real-world adversarial patches may not always be square, the robust transferability of the model to shapes other than the square is important.

Therefore, we evaluate the robustness of the square-patch-trained model to adversarial patches of different shapes while fixing the number of pixels. In all these experiments, we evaluate the certified accuracy for our largest model, on both MNIST and CIFAR datasets. We evaluate the transferability to various shapes including rectangle, line, parallelogram, and diamond. With the exception of rectangles, all the shapes have the exact same pixel count as the patches used for training. For rectangles, we use multiple choices of width and length, obtaining some combinations with slightly more pixels, and the worst accuracy is reported in Table 5. The exact shapes used can be found in Appendix A.2.

The certified accuracy of our models generalize surprisingly well to other shapes, losing no more than than 5% in most cases for MNIST and no more than 6% for CIFAR-10 (Table 5). The largest degradation of accuracy happens for rectangles and lines, and it is mostly because the rectangle considered has more pixels compared to the square, and the line has less overlaps. However, it is still interesting that the certificate even generalizes to a straight line, even though the model was never trained to be robust to lines. In the case of MNIST with small patch size, the certified accuracy even improves when transferred to lines.

Table 5: Certified accuracy for square-patch trained model for different shapes

| Dataset | Pixel Count | Square | Rectangle | Line | Diamond | Parallelogram |
|---|---|---|---|---|---|---|
| MNIST | 4 | 91.6% | - | 92.5% | 91.6% | 92.3% |
| | 16 | 69.4% | 55.4% | 46.7% | 68.13% | 70.2% |
| | 25 | 59.7% | 50.9% | 32.4% | 53.6% | 55.2% |
| CIFAR | 4 | 50.8% | - | 46.1% | 48.6% | 49.8% |
| | 16 | 36.9% | 29.0% | 32.1% | 35.7% | 36.3% |
| | 25 | 30.3% | 25.1% | 29.0% | 30.1% | 30.7% |

## 6 CONCLUSION AND FUTURE WORK

After establishing the weakness of known defenses to patch attacks, we proposed the first certified defense against this model. We demonstrated the effectiveness of our defense on two datasets, and proposed strategies to speed up robust training. Finally, we established the robust transferability of trained certified models to different shapes. In its current form, the proposed certified defense is unlikely to scale to ImageNet, and we hope the presented experiments will encourage further work along this direction.

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

# A APPENDIX

## A.1 EXPERIMENTAL SETTINGS AND NETWORK STRUCTURE

We evaluate the proposed certified patch defense on three neural networks: a multilayer perceptron (MLP) with one 255-neuron hidden layer, and two convolutional neural networks (CNN) with different depths. The small CNN has two convolutional layers (kernel size 4, stride 2) of 4 and 8 output channels each, and a fully connected layer with 256 neurons. The large CNN has four convolutional layers with kernel size (3, 4, 3, 4), stride (1, 2, 1, 2), output channels (4, 4, 8 ,8), and two fully connected layer with 256 neurons. We run experiments on two datasets, MNIST and CIFAR10, with two different patch sizes $2 \times 2$ and $5 \times 5$. For all experiments, we are using Adam (Kingma & Ba, 2014) with a learning rate of $5e - 4$ for MNIST and $1e - 3$ for CIFAR10, and with no weight decay. We also adopt a warm-up schedule in all experiments like (Zhang et al., 2019b), where the input interval bounds start at zero and grow to [-1,1] after 61/121 epochs for MNIST/CIFAR10 respectively. We train the models for a total of 100/200 epochs for MNIST/CIFAR10, where in the first 61/121 epochs the learning rate is fixed and in the following epochs, we reduce the learning rate by one half every 10 epochs.

In addition, following (Gowal et al., 2018), we further evaluate the CIFAR10 on a larger model which has 5 convolutional layers with kernel size 3 and a fully connected layer with 512 neurons. This deeper and wider model achieves the clean accuracy around $89\%$, and has 17M parameters in total. Table 8 in Appendix A.5 describes the full certified patch results for this large model.

## A.2 SAMPLE SHAPES FOR GENERALIZATION EXPERIMENTS

We demonstrate generalization to other patch shapes that were not considered in training, obtaining surprisingly good transfer in robust accuracy; see the figure below and the results in Table 5.

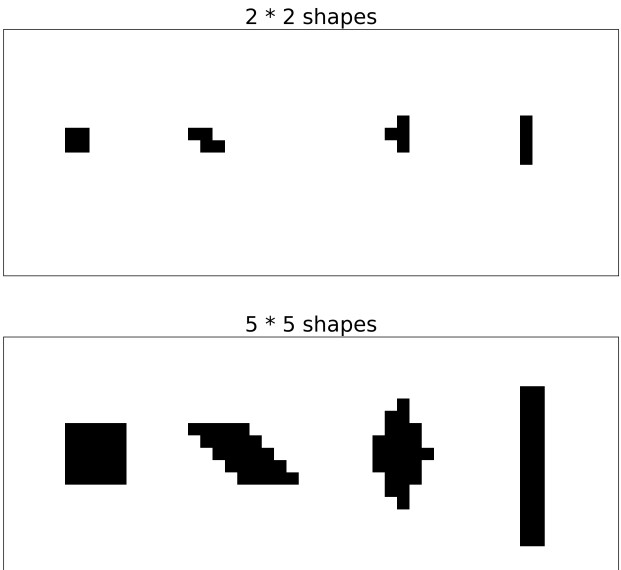

Figure 1: Examples of shapes with pixels number 4 and 25. From left to right are square, parallelogram, diamond and rectangle (line) respectively.

## A.3 BOUND POOLING

Besides random-patch certificate training and guided-patch certificate training, we also experimented with the idea of bound pooling. All-patch training is very expensive as bounds generated by each potential patch has to be forward passed through the *complete* network. Bound pooling partially relieves the problem be pooling the interval bounds in intermediate layers thus reducing the forward pass in subsequent layers.

Specifically, given a set of patches $\mathbb{P}$, the interval bounds in the $i$th layer are $\bar{Z}^{(i)}(\mathbb{P}) = \{\bar{z}^{(i)}(p) : p \in \mathbb{P}\}$ and $\underline{Z}^{(i)}\mathbb{P} = \{\underline{z}^{(i)}(p) : p \in \mathbb{P}\}$. We can reduce the number of interval bounds by partitioning $\mathbb{P}$ into $n$ subsets $\{\mathbb{S}^1, ..., \mathbb{S}^n\}$ and calculate a new set of bounds $\bar{Z}^{(i)}_{pool}(\mathbb{P}) = \{\max_{p \in \mathbb{S}_i} \bar{z}^{(i)}(p) : i \in [n]\}$ and $\underline{Z}^{(i)}_{pool}(\mathbb{P}) = \{\min_{p \in \mathbb{S}_i} \underline{z}^{(i)}(p) : i \in [n]\}$. Depending on how $\mathbb{P}$ is partitioned, the bound pooling would work differently. In our experiments, we always select adjacent patches for each $\mathbb{S}_i$ with the assumption that adjacent patches tend to generate similar bounds thus resulting in tighter certificate.

Bound pooling, similar to random- and guided- patch training, trades performance for efficiency compared to all-patch certificate training. However, the trade off is not as favorable compared to random-patch and guided-patch training. For example, in Table 6, Pooling 16 ($4 \times 4$) patches in the first layer reduces training time by 35% while loosing 0.7% in performance (on MNIST $2 \times 2$), but a similar level of performance can be achieved with guided-patch training with almost 90% reduction in training time. The trade off becomes greater when the model becomes larger. Overall, bound pooling is still quite expensive, and cannot scale to larger models like random-patch or guided-patch training.

Table 6: Comparing bound pooling with the guided-patch and random-patch training. Pool 4 means that the adjacent $4 \times 4$ patches (16 patches) are pooled together in the first layer. Pool 2-2 means that the adjacent $2 \times 2$ bounds are pooled together in the first layer and then another $2 \times 2$ bound pooling happens at the second layer. This is similar to $4 \times 4$ pooling except the pooling operation is distributed between the first and second layer. All experiments are performed on a 4-layer convolutional network.

| Dataset | Training Strategy | $2 \times 2$ | | | $5 \times 5$ | | |
|---|---|---|---|---|---|---|---|
| | | Clean Accuracy | Certified Accuracy | Training Time(h) | Clean Accuracy | Certified Accuracy | Training Time(h) |
| MNIST | All Patch | 98.5% | 91.6% | 20.1 | 90.0% | 59.7% | 16.3 |
| | Pool 2 | 98.0% | 91.1% | 15.8 | 85.2% | 54.2% | 11.6 |
| | Pool 4 | 97.2% | 89.9% | 13.2 | 70.4% | 38.3% | 10.2 |
| | Random(1) | 98.5% | 81.9% | 0.3 | 96.8% | 24.8% | 0.4 |
| | Random(5) | 98.6% | 86.5% | 0.3 | 94.9% | 42.0% | 0.5 |
| | Random(10) | 98.6% | 87.5% | 0.5 | 94.7% | 50.4% | 0.6 |
| | Guided(10) | 98.7% | 88.9% | 2.2 | 94.0% | 53.2% | 3.4 |
| CIFAR | All Patch | 49.6% | 41.6% | 22.5 | 34.0% | 25.0% | 18.6 |
| | Pool 2 | 48.1% | 39.4% | 17.3 | 32.4% | 24.2% | 14.5 |
| | Pool 4 | 44.9% | 37.1% | 16.3 | 28.3% | 20.6% | 13.6 |
| | Pool 2-2 | 45.0% | 37.4% | 16.5 | 25.3% | 19.1% | 13.8 |
| | Random(1) | 53.2% | 32.4% | 0.6 | 42.7% | 11.0% | 0.6 |
| | Random(5) | 52.2% | 39.5% | 0.9 | 37.8% | 19.6% | 0.9 |
| | Random(10) | 50.8% | 38.6% | 1.0 | 38.4% | 21.9% | 1.0 |
| | Guided(10) | 53.0% | 39.8% | 4.0 | 36.1% | 23.0% | 3.9 |

### A.4 MULTI-PATCH SPARSE TRAINING

Here we list the detailed certified accuracy for various sparsity levels and model architectures.

Table 7: Certified accuracy for sparse defenses with varying sparsity $k$ and models on both MNIST and CIFAR10, where "Conv $c \times c$" represents for the convolutional network with first layer kernel size $c$.

| Dataset | Sparsity ($k$) | Model | Clean Accuracy | Certified Accuracy |
|---|---|---|---|---|
| MNIST | 1 | mlp | 98.4% | 96.0% |
| | 4 | mlp | 97.8% | 90.8% |
| | 10 | mlp | 95.2% | 86.8% |
| | 1 | Conv3x3 | 97.0% | 88.3% |
| | 4 | Conv3x3 | 92.7% | 75.9% |
| CIFAR | 1 | mlp | 48.4% | 40.0% |
| | 4 | mlp | 42.2% | 31.2% |
| | 10 | mlp | 37.0% | 25.6% |
| | 1 | Conv11x11 | 34.8% | 27.4% |
| | 4 | Conv11x11 | 25.1% | 18.3% |
| | 10 | Conv11x11 | 17.2% | 13.8% |
| | 1 | Conv13x13 | 38.6% | 29.7% |
| | 4 | Conv13x13 | 28.1% | 19.6% |
| | 10 | Conv13x13 | 22.4% | 15.3% |

### A.5 TRAINING WITH LARGER MODELS

Recall that all-patch training considers all possible patches during training, which can be too expensive for larger models and/or images. The proposed random- and guided-patch training methods aim to reduce the training cost by considering only a subset of patches; please see Section 4.2 for more details.

Table 8: The random and guided training strategy could yield significantly stronger model compared to all-patch training given a fixed computational budget. The random and guided training strategy allows us to train a larger model that would be infeasible to train otherwise. The guided-patch large model is able to boost the certified accuracy by over 10% compared to the best performing all-patch small model.

| Dataset | Patch Size | Training Strategy | Model | Clean Accuracy | Certified Accuracy | Training Time(h) |
|---|---|---|---|---|---|---|
| CIFAR | $2\times 2$ | All Patch | mlp | 50.8% | 35.5% | 9.1 |
| | | | 2 layer conv | 52.4% | 42.6% | 10.7 |
| | | | 4 layer conv | 49.6% | 41.6% | 22.5 |
| | | | 5 layer conv (wide) | - | - | ~360.0 |
| | | Random(10) | 5 layer conv (wide) | 64.7% | 49.0% | 9.5 |
| | | Random(20) | 5 layer conv (wide) | 64.4% | 50.8% | 15.8 |
| | | Guided(10) | 5 layer conv (wide) | **66.5%** | 49.2% | 12.2 |
| | | Guided(20) | 5 layer conv (wide) | 65.8% | **51.9%** | 22.4 |
| CIFAR | $5\times 5$ | All Patch | mlp | 31.1% | 18.8% | 7.1 |
| | | | 2 layer conv | 35.5% | 22.3% | 8.7 |
| | | | 4 layer conv | 34.0% | 25.0% | 18.6 |
| | | | 5 layer conv (wide) | - | - | ~360.0 |
| | | Random(10) | 5 layer conv (wide) | **48.6%** | 29.9% | 9.4 |
| | | Random(20) | 5 layer conv (wide) | 47.8% | **30.3%** | 15.4 |
| | | Guided(10) | 5 layer conv (wide) | 48.4% | 29.0% | 12.4 |
| | | Guided(20) | 5 layer conv (wide) | 47.6% | 29.6% | 23.8 |

## A.6  Detailed Statistics on Training Strategies

Here we list the detailed statistics for each training strategies for Table 3

| Dataset | Training Strategies | Model Architecture | Clean Accuracy | Certified Accuracy | Training Time |
|---|---|---|---|---|---|
| MNIST | All Patch | 2 layer convolution | 98.63/% | 91.38% | 21.0 |
| | | 4 layer convolution | 98.48% | 91.63% | 80.3 |
| | | fully connected (255,10) | 98.46% | 91.47% | 9.8 |
| | Random (1) | 2 layer convolution | 98.69% | 82.57% | 0.2 |
| | | 4 layer convolution | 98.45% | 81.87% | 0.3 |
| | | fully connected (255,10) | 98.48% | 84.32% | 0.2 |
| | Random (5) | 2 layer convolution | 98.75% | 85.87% | 0.3 |
| | | 4 layer convolution | 98.57% | 86.50% | 0.3 |
| | | fully connected (255,10) | 98.62% | 87.32% | 0.2 |
| | Random (10) | 2 layer convolution | 98.73% | 87.31% | 0.3 |
| | | 4 layer convolution | 98.63% | 87.54% | 0.5 |
| | | fully connected (255,10) | 98.49% | 88.13% | 0.2 |
| | Guided (10) | 2 layer convolution | 98.60% | 88.49% | 2.3 |
| | | 4 layer convolution | 98.70% | 88.85% | 2.2 |
| | | fully connected (255,10) | 98.63% | 89.44% | 2.2 |
| CIFAR | All Patch | 2 layer convolution | 52.42% | 42.57% | 42.6 |
| | | 4 layer convolution | 49.58% | 41.57% | 89.8 |
| | | fully connected (255,10) | 50.83% | 35.49% | 36.6 |
| | Random (1) | 2 layer convolution | 54.93% | 29.13% | 0.6 |
| | | 4 layer convolution | 53.22% | 32.35% | 0.6 |
| | | fully connected (255,10) | 52.76% | 03.21% | 0.5 |
| | Random (5) | 2 layer convolution | 54.15% | 37.30% | 0.6 |
| | | 4 layer convolution | 52.19% | 39.45% | 0.9 |
| | | fully connected (255,10) | 52.38% | 20.17% | 0.6 |
| | Random (10) | 2 layer convolution | 53.08% | 39.32% | 0.7 |
| | | 4 layer convolution | 50.80% | 38.57% | 1.0 |
| | | fully connected (255,10) | 51.90% | 28.97% | 0.6 |
| | Guided (10) | 2 layer convolution | 53.04% | 38.81% | 3.7 |
| | | 4 layer convolution | 52.97% | 39.84% | 4.0 |
| | | fully connected (255,10) | 51.32% | 29.44% | 3.6 |

Table 9: Detailed statistics for the comparison of training strategies - 2×2

| Dataset | Training Strategies | Model Architecture | Clean Accuracy | Certified Accuracy | Training Time |
|---------|---------------------|--------------------|----------------|--------------------|---------------|
| MNIST | All Patch | 2 layer convolution | 91.88% | 59.59% | 28.4 |
| | | 4 layer convolution | 90.03% | 59.72% | 65.2 |
| | | fully connected (255,10) | 93.96% | 61.97% | 7.2 |
| | Random (1) | 2 layer convolution | 96.27% | 18.57% | 0.2 |
| | | 4 layer convolution | 96.83% | 24.79% | 0.4 |
| | | fully connected (255,10) | 97.60% | 29.04% | 0.2 |
| | Random (5) | 2 layer convolution | 95.82% | 38.47% | 0.2 |
| | | 4 layer convolution | 94.85% | 42.02% | 0.5 |
| | | fully connected (255,10) | 96.73% | 45.89% | 0.2 |
| | Random (10) | 2 layer convolution | 95.55% | 46.13% | 0.3 |
| | | 4 layer convolution | 94.76% | 50.43% | 0.6 |
| | | fully connected (255,10) | 96.40% | 52.30% | 0.2 |
| | Guided (10) | 2 layer convolution | 95.28% | 50.28% | 2.3 |
| | | 4 layer convolution | 93.98% | 53.17% | 3.4 |
| | | fully connected (255,10) | 95.82% | 55.89% | 2.2 |
| CIFAR | All Patch | 2 layer convolution | 35.48% | 22.31% | 34.8 |
| | | 4 layer convolution | 33.95% | 24.96% | 74.4 |
| | | fully connected (255,10) | 31.05% | 18.78% | 28.4 |
| | Random (1) | 2 layer convolution | 45.71% | 07.14% | 0.6 |
| | | 4 layer convolution | 42.65% | 10.99% | 0.6 |
| | | fully connected (255,10) | 42.34% | 00.10% | 0.5 |
| | Random (5) | 2 layer convolution | 42.85% | 17.29% | 0.6 |
| | | 4 layer convolution | 37.80% | 19.63% | 0.9 |
| | | fully connected (255,10) | 36.23% | 06.99% | 0.6 |
| | Random (10) | 2 layer convolution | 41.90% | 21.40% | 0.7 |
| | | 4 layer convolution | 38.41% | 21.90% | 1.0 |
| | | fully connected (255,10) | 36.04% | 12.46% | 0.6 |
| | Guided (10) | 2 layer convolution | 42.08% | 20.77% | 3.6 |
| | | 4 layer convolution | 36.08% | 23.01% | 3.9 |
| | | fully connected (255,10) | 35.51% | 12.56% | 3.5 |

Table 10: Detailed statistics for the comparison of training strategies - 5×5

