# OpenReview forum: "Certified Defenses for Adversarial Patches"
_ICLR.cc/2020/Conference — Accept (Poster)_

### Official Review · AnonReviewer3 · 2019-10-22
**Official Blind Review #3**

**Rating:** 6

**Review:**

This paper proposed a certified defense method for adversarial patches. The paper is motivated by the finding that several existing works on adversarial patch defenses are easily "breakable" by in the white-box setting. The idea of the proposed method is derived Interval Bound Propagation (IBP), which is originally proposed for certified defense against adversarial noise. To simplify the certificate training in the patch defense setting (which original scales quadratically with respect to the image size), two randomized training methods are proposed. Lastly, experimental results indeed verify the effectiveness of the proposed method.

The paper is well-written and very well-organized. It is interesting to see supposedly strong adversarial patch defense methods "break down" in a very simple setting. And the contribution of the paper is significant to the field.

I do have a question with regard to the randomized training method:
Although random patches (or selected worse-case patches at random location) can be used for certificate training, in order to create a certificate during testing, does it mean that you do have to conduct many forward pass of the network with respect to all patches at all locations?

**Experience Assessment:**

I have read many papers in this area.

**Review Assessment: Checking Correctness Of Derivations And Theory:**

I assessed the sensibility of the derivations and theory.

**Review Assessment: Checking Correctness Of Experiments:**

I assessed the sensibility of the experiments.

**Review Assessment: Thoroughness In Paper Reading:**

I read the paper at least twice and used my best judgement in assessing the paper.

---

> ### Author Response · Authors · 2019-11-14
> **Response to Reviewer #3**
>
> Thanks for the positive assessment and supportive remarks!
>
> R3: “have to conduct many forward pass of the network with respect to all patches at all locations?”
>
> Author(s): Yes, we forward pass all patches in order to get a certificate. We have included an additional paragraph in section 4.2 to clarify.
>
> ***Other Comments***
> - There was a minor bug in our code that didn’t have a significant impact on any of the results. All tables were updated with new runs.

---

### Official Review · AnonReviewer2 · 2019-10-22
**Official Blind Review #2**

**Rating:** 6

**Review:**

The paper proposes a certified defense for adversarial patch attacks.
Technically, the authors use the well developed IBP based methods (Gowal et
al., 2018, Mirman et al., 2018, Zhang et al. 2019).  The technique is simple
but effective. Since the number of possible patches are quadratic w.r.t image
dimension, to reduce the number of bounds to propagate, the authors propose a
U-Net based NN to predict the worst case scenario, and only propagate "worst
case" bounds predicted by the U-net.

Empirically, the proposed method gets good results, with certified accuracy
sometimes even higher than empirical accuracy by previous methods.  The authors
also provide results for transferring robustness properties to shapes that are
not included during training.

Overall, the contribution of this paper is novel, and results are promising,
but it still has some missing components, especially the idea of combining
multiple IBP bounds into one, which can be very effective for adversarial
patches, as I will elaborate below.

Suggestions and Questions:

The core idea behind IBP is that for whatever input perturbation is given (any
Lp norm or semi-norm, or non-norm based perturbations like patches at arbitrary
locations), it converts them to per-neuron lower and upper bounds after the
first linear/conv layer.  For example, if the input perturbation is *two* patches
B_1 and B_2, after propagating them through the first layer of the network, we
got two lower bounds l_{i,1}, l_{i,2} and two upper bounds u_{i,1}, u_{i,2} for
the i-th neuron. We then take the worst case bound, l_i = min(l_{i,1},
l_{i,2}), u_i = max(u_{i,1}, u_{i,2}) and propagate only one set of bounds l_i
and u_i to the next layer. The authors should explore on this direction, as
detailed below:

1. For the exhaustive patch enumeration in (11), we can actually greatly reduce
the computation cost by combining the bounds of different patches after the
first layer of the network, as I mentioned above. At the input layer, the
number of bounds (each for one possible location of patch) are large; but after
the first linear/conv layer, we can compress them to one or a small group of
bounds by taking the worst cast of them, like l_i = min(l_{i,1}, ...,
l_{i,|L|}), u_i = max(u_{i,1}, ..., u_{i,2}). The patches close with each other
should also produces similar lower and upper bounds, so taking min or max over
them will not make the combined bounds much worse. This is better than U-net
prediction since we are guaranteed to include the worst case scenario.

2. Considering multiple patches (at different locations) on a single input. In
the simplest case, consider multiple 1x1 patches (in fact, this is equivalent
to bounded L0 norm threat model); since each patch is 1x1 (only changing one
pixel), the bounds should be relatively tight after the first layer, and after
the first linear/conv layer we got 28*28 or 32*32 bounds which will be combined
into one or very few sets of lower and upper bounds that will be propagate into
later layers.  Multiple larger patches (2x2, 5x5) can be difficult since bounds
are looser; multiple 1x1 in my opinion is both technically feasible and
practically important, and should definitely be included in this paper.


Minor issues:

1. Eq. (5) and (6) are incorrect; the second term should be \overline{z} -
\underline{z}. Also it is missing the bias term.

2. In related works (section 4.1, page 3), (Weng et al., 2018) is not a defense
method (it is a certification method, and no training is involved).

3. In Table 3, it is better if the authors can provide empirical adversarial
accuracy to IBP defended networks as well.

Overall, I think it is a good paper but the authors should explore more to
strengthen their contributions. I gave a weak reject but I will not hesitate
to recommend an accept as long as the authors can provide additional results
mentioned above.


**Experience Assessment:**

I have published in this field for several years.

**Review Assessment: Checking Correctness Of Derivations And Theory:**

I carefully checked the derivations and theory.

**Review Assessment: Checking Correctness Of Experiments:**

I assessed the sensibility of the experiments.

**Review Assessment: Thoroughness In Paper Reading:**

I read the paper thoroughly.

---

> ### Author Response · Authors · 2019-11-14
> **Response to Reviewer #2**
>
> Thanks for the support and valuable suggestions!
>
> *** Main Comments ***
>
> R2: “still has some missing components, especially the idea of combining multiple IBP bounds into one [..] we can actually greatly reduce the computation cost by combining the bounds of different patches after the first layer of the network. [..] This is better than U-net prediction ..”
>
> Author(s): The proposed bound pooling still scales quadratically w.r.t. image dimension, although it does reduce computational costs compared to the straightforward all-patch training. In contrast, the proposed random- and guided-patch approaches scale better allowing the training of larger models.
>
> A new Appendix A.3 includes extra experiments with bound pooling: We tested bound pooling over 2⨉2/4⨉4 patches in the first layer, 2⨉2 patches in the first layer and 4⨉4 patches in the second layer. Compared to all-patch training, this implementation of bound pooling indeed reduces the computational cost by 25-35% while being only slightly worse in terms of certified accuracy; please refer to Table 5.
>
>
> R2: “Considering multiple patches (at different locations) on a single input. [..] multiple 1x1 in my opinion is both technically feasible and practically important, and should definitely be included in the paper”
>
> Author(s): Unfortunately, this interesting setup is computationally infeasible. We actually considered this formulation earlier but could not pursue it due to the combinatorial explosion in the number of patches, see below for more details.
>
> In the specific example of 1⨉1 patch, when we take the maximum of upper bounds generated by all patches, we are not getting the upper bounds for moving all pixels at the same time, rather we get the worst case bounds when we are allowed to move a single pixel, but not moving any of them together. To give an actual certificate, we have to consider (32⨉32) position for the first patch and (32⨉32-1) position for the second patch, and we would have to forward pass all of these possible combinations forward in order to get an actual certificate. Unfortunately, this is computationally infeasible, but is very interesting nonetheless.
>
>
> *** Other Comments ***
> - We have removed reference to Weng's
> - We have fixed the typos in equation (5) & (6)
> - We have included empirical adversarial accuracy to Table 3
> - There was a minor bug in our code that didn’t have a significant impact on any of the results. All tables were updated with new runs.

---

> > ### Comment · AnonReviewer2 · 2019-11-15
> > **Thanks for the additional results; I am considering increasing the rating; however I believe multiple 1x1 boxes are doable**
> >
> > Thanks for the additional results on combining multiple boxes. Please make sure to mention it in the main text for the final revision, as I believe it is an important contribution.
> >
> > I am considering increasing the rating. However I believe multiple 1x1 boxes are doable.
> >
> > Let consider there are 32x32 positions (so dimension of input $x$ 1024), and we have a linear layer with weights $W$ in shape $N \times 1024$, where $N$ is the number of neurons. For each neuron, we can see its output as a dot product between the 1024-dimensional input, and a row of the weight matrix (denoted as $w$). Now we consider the case, where we can only perturb 1 of the 32x32 positions. What is the output lower bound and upper bound for that neuron? Apparently, we can increase the output of that neuron by increasing a pixel value that is corresponding to a positive coefficient in $w$, and vise versa. And apparently to reached the upper and lower bound, we perturb the single pixel corresponding to largest element in $w$. Importantly, you just need to look at the largest element in $w$ to determine its output upper and lower bound. We don't need 32x32 boxes at all.
> >
> > Now we consider the general case, where we can perturb $k$ of the 32x32 positions. Then you need to look into the $k$ largest element in $w$ to determine the neurons output lower and upper bounds. There is no need to construct 32x32 boxes, again.
> >
> > I strongly encourage the authors to look into the multiple 1x1 patch case above, as it is a very important use case, and also should be fairly straightforward and efficient to implement.

---

> > > ### Author Response · Authors · 2019-11-15
> > > **Thanks for point this out!**
> > >
> > > Thanks for following up. We were able to get new results for multiple 1x1 patches as suggested, and will include them in the manuscript upon acceptance.
> > >
> > > R2: "Please make sure to mention it in the main text for the final revision, as I believe it is an important contribution. "
> > >
> > > Author(s): Yes, we will make sure to include it in the manuscript upon acceptance.
> > >
> > > R2: "However I believe multiple 1x1 boxes are doable."
> > >
> > > Author(s): Indeed, you are correct. Thanks for pointing this out. We have included results for both MNIST & CIFAR for the 1,4,10, 25 1x1 patches below. We will also update this in the main text upon acceptance.
> > >
> > > ╔═════╦══╦════════╦═══════════╦═════════════╗
> > > ║              ║ k  ║ Model          ║ Clean Accuracy  ║ Certified Accuracy  ║
> > > ╠═════╬══╬════════╬═══════════╬═════════════╣
> > > ║ MNIST ║ 1  ║ MLP             ║ 98.4%                   ║  96.0%                      ║
> > > ╠═════╬══╬════════╬═══════════╬═════════════╣
> > > ║ MNIST ║ 4  ║ MLP             ║ 97.8%                   ║ 90.8%                       ║
> > > ╠═════╬══╬════════╬═══════════╬═════════════╣
> > > ║ MNIST ║ 10║ MLP             ║ 95.2%                  ║ 86.8%                        ║
> > > ╠═════╬══╬════════╬═══════════╬═════════════╣
> > > ║ MNSIT ║ 25║ MLP             ║ 68.2%                  ║ 35.6%                        ║
> > > ╠═════╬══╬════════╬═══════════╬═════════════╣
> > > ║ MNIST ║ 1  ║ Conv3x3      ║ 97.0%                  ║ 88.3%                        ║
> > > ╠═════╬══╬════════╬═══════════╬═════════════╣
> > > ║ MNIST ║ 4  ║ Conv3x3      ║ 92.7%                  ║ 75.9%                        ║
> > > ╠═════╬══╬════════╬═══════════╬═════════════╣
> > > ║ CIFAR  ║ 1  ║ MLP              ║ 48.4%                  ║ 40.0%                        ║
> > > ╠═════╬══╬════════╬═══════════╬═════════════╣
> > > ║ CIFAR  ║ 4  ║ MLP              ║ 42.2%                  ║ 31.2%                        ║
> > > ╠═════╬══╬════════╬═══════════╬═════════════╣
> > > ║ CIFAR  ║ 10 ║ MLP             ║ 37.0%                  ║ 25.6%                        ║
> > > ╠═════╬══╬════════╬═══════════╬═════════════╣
> > > ║ CIFAR  ║ 1  ║ Conv11x11  ║ 34.8%                  ║ 27.4%                        ║
> > > ╠═════╬══╬════════╬═══════════╬═════════════╣
> > > ║ CIFAR  ║ 4  ║ Conv11x11  ║ 25.1%                  ║ 18.3%                        ║
> > > ╠═════╬══╬════════╬═══════════╬═════════════╣
> > > ║ CIFAR  ║ 10 ║ Conv11x11 ║ 17.2%                  ║ 13.8%                        ║
> > > ╠═════╬══╬════════╬═══════════╬═════════════╣
> > > ║ CIFAR  ║ 1  ║ Conv13x13  ║ 38.6%                  ║ 29.7%                        ║
> > > ╠═════╬══╬════════╬═══════════╬═════════════╣
> > > ║ CIFAR  ║ 4   ║ Conv13x13 ║ 28.1%                  ║ 19.6%                        ║
> > > ╚═════╩══╩════════╩═══════════╩═════════════╝
> > >
> > > It is interesting to note that fully connected only architecture performs the best among all architectures that we experimented in the past few hours. The approach does not perform very well when applied to the convolutional networks because the matrix is very sparse leading to very loose interval bounds with the top-k approach. Moreover, the interval bounds would be vacuous when the number of pixels k is larger than the size of the convolutional kernel. It seems like increasing the size of the convolutional kernel (in the first layer only) improves the performance, but still does not match that of the fully connected. We think it is a very interesting direction to adapt the L0 interval bound to work better with the convolutional layer, and we will further investigate this in our future work.
> > >
> > > >>> We will also add the method and results in the manuscript upon acceptance.

---

> > > > ### Comment · AnonReviewer2 · 2019-11-15
> > > > **Thanks for the encouraging results**
> > > >
> > > > Thanks for providing the encouraging initial results. Make sure to formally formulate the bound in this case and include these results in the final version of the paper.
> > > >
> > > > A caveat is that the input is bounded (e.g., for MNIST the image is in range 0 to 1). For example, if a pixel is already close to 1, it cannot be perturbed up even its corresponding element $w$ is large. A more careful approach will consider both the magnitude of elements in $w$, and the available perturbation budget. But it still only needs to check the elements in $w$ and the input bounds once, so does not increase time complexity.
> > > >
> > > > Yes I also believe that this bound can be relatively loose for convolutional layers - especially when $k$ is large, e.g., $k$ equals to the number of non-zero elements in $w$, then the perturbation is like a L infinity perturbation with unbounded $\epsilon$. For small $k$ the provided results look reasonable. Some discussions on this can be useful in the paper.
> > > >
> > > > I have changed my rating to weak accept, and tend to accept this paper based on the new results.

---

> > > > > ### Author Response · Authors · 2019-11-15
> > > > > **Thank you!**
> > > > >
> > > > > Thank you for supporting our paper!
> > > > >
> > > > > >>> We will make sure to formulate the bounds for this case more formally, and include that in the manuscript.

---

### Official Review · AnonReviewer1 · 2019-10-23
**Official Blind Review #1**

**Rating:** 6

**Review:**

This paper attempts to extend the Interval Bound Propagation algorithm from (Gowal et al. 2018) to defend against adversarial patch-based attacks. In order to defend against patches which could appear at any location, all the patches need to be considered. This is too computationally expensive, hence they proposed to use a random subset of patches, or a U-net to predict the locations of the patches and then use those patches to train. The algorithm is tested on the MNIST and CIFAR-10 datasets and it was shown that sometimes the IBP approach is useful for defense, although often with a significant loss on accuracy on clean data (e.g. on CIFAR the loss on clean accuracy is an astounding 300% -- from 66.5% - 35.7%).

I think the technical contribution of this paper is a bit weak in that they mostly followed the original IBP and the only novelties are the random patch training and guided patch training. I partially like how the experiments are conducted, especially the one that generalizes to other shapes. On the other hand, the networks that are tested seem pretty poor by any standard. An experiment that is definitely missing is a CIFAR network that performs a little better than the current one. Clean accuracy of only 66.5% and 47.2% are very lousy for CIFAR.

Another missing experiment is one that would test on different epsilon values. I couldn't find what are the current epsilon values used?

Besides, since this work is testing on adversarial patches, I would like to at least have it applied to some real-life images with patches that are of real-life size. I could care a bit less on how good it is, but one can still make an empirical test (e.g. certified defense accuracy on 5x5 patches, but empirical test using real-life sized patches 40x40 or 80x80) and see how the results would be. All the experiments mentioned above would significantly strengthen the experiments section of the paper.

I don't think I read anywhere a confirmation that the testing is performed on all patches of the prescribed size. Could the authors please confirm whether this is true?

Minor:
There is a typo in Eq. (5) and Eq. (6), where the second term multiplied by |W^(k)| should be \underline{z}^(k-1) - \bar{z}^(k-1) instead of \underline{z}^(k-1) + \bar{z}^(k-1)

You should mention that |W^(k)| stand for element-wise absolute value when it first appears.

**Experience Assessment:**

I have published one or two papers in this area.

**Review Assessment: Checking Correctness Of Derivations And Theory:**

I carefully checked the derivations and theory.

**Review Assessment: Checking Correctness Of Experiments:**

I carefully checked the experiments.

**Review Assessment: Thoroughness In Paper Reading:**

I read the paper thoroughly.

---

> ### Author Response · Authors · 2019-11-14
> **Response to Reviewer 1 (2/2)**
>
> Continued ..
>
> R1: “contribution of the paper is a bit weak in that they mostly followed the original IBP and the only novelties are the random patch training and guided patch training”
>
> Our contributions are a mix of technical advancements and empirical improvements.
>
> Our framework is indeed based off the IBP concept, however the proposed adaptation to patch attacks (a more realistic attack model in many applications) is non-trivial, and we explore three different strategies towards practical adversarial patch training. Compared to the straightforward all-patch defense, under a fixed computational budget, the accuracy of the proposed random-patch training and guided-patch training is significantly improved, which is not necessarily obvious or expected. Even the seemingly interesting idea of incorporating bound pooling after the first layer, as proposed by the expert Reviewer #2, did not reduce computation enough to scale to larger models.
>
> We also make a range of empirical observations that we think are important enough to count as contributions. Namely, we show that state-of-the-art empirical patch defenses are easily breakable, and quantify this effect.
>
> Finally, the clean and robust accuracies we achieve exceed other L-infinity certified models and existing patch defense methods. As such, we believe that our work makes an important step towards patch defenses.
>
> R1: “real-life images with patches that are of real-life size”
>
> Author(s): As CIFAR images are only 32⨉32, we are unable to experiment with patch size this big. If we want to do certification for 40x40, we would have to conduct our experiments on ImageNet. As it currently stands, IBP has difficulty extending to imagenet sized datasets, and we leave this as future work. Note that CIFAR-10 is currently a widely used benchmark for adversarial defense (both empirical and certified).
>
> *** Other Comments ***
> - We have fixed the typos in equation (5) & (6)
> - Yes, we forward pass all patches in order to get a certificate for all of our experiments. We have included an additional paragraph in section 4.2 to clarify.
> - Our lower and upper bound is simply the image domain (0-1), and we don't really have an epsilon.
> - We added description for the element-wise absolute value
> - There was a minor bug in our code that didn’t have a significant impact on any of the results. All tables were updated with new runs.

---

> ### Author Response · Authors · 2019-11-14
> **Response to Reviewer 1 (1/2)**
>
> Thanks for the careful remarks with a healthy dose of skepticism that helped improve the manuscript. We ran additional experiments using a larger model, as requested, and elaborated on the significance of the reported results w.r.t. state-of-the-art in certified accuracy.
>
> *** Main Comments ***
>
> R1: “An experiment that is definitely missing is a CIFAR network that performs a little better”
>
> Author(s): Per your suggestion, in the revised manuscript, we have trained a new model which performs considerably better in terms of validation accuracy on benign examples. The new defended model which is a considerably larger model achieves 66.5% accuracy* on the benign examples (a 16.2% increase compared to the narrower convolutional models in the original submission). The certified accuracy for the new model on the CIFAR 2x2 task is also improved to 51.9% (a 10.3% increase). The large model significantly outperforms the previous all-patch trained model, even though it has only been trained using guided patches. Note that we were not able to complete all-patch training during the rebuttal phase on the large model as it takes almost 15 days to complete the training on our workstation.
>
> >>> We have updated our manuscript to include the experimental results for the large model in Appendix A.4. Upon acceptance, we will include all-patch results in the camera-ready version.
>
> (*) For the non-defended model, the large model achieve 88.9% clean accuracy.
>
>
> R1: “sometimes the IBP approach is useful for defense, although often with a significant loss in accuracy on clean data”
>
> Author(s): The trade-off between robustness and generalization has been argued to exist both from a theoretical perspective and also empirically [3, 4]. Some loss in accuracy in certified robustness compared to clean accuracy is typical for all certified (and most empirical) defenses. Our proposed certified defense, which is the first certified defense against adversarial patch attacks, is no exception to this. To put our reported results in perspective, we consider state-of-the-art results for (a) certified robustness against L-inf perturbations, and (b) (non-certifying) empirical defenses against patch attacks; see below.
>
> (a) The State-of-the-art certified model for CIFAR with 16/255 L-inf perturbation using deterministic algorithms [5] achieves clean accuracy of 34.0%, whereas our clean accuracy for our most robust CIFAR 5 ⨉ 5 model is 47.8% (for the updated large model). To further appreciate the performance of our model, note that our threat model allows the attacker to change each pixel within the patch to any desired value, whereas the L-inf perturbation model of [5] assumes the adversary can only change each pixel by at most some epsilon, e.g., 16/255.
>
> >>> This remark was added to the manuscript at the bottom of Section 5.1.
>
> (b) State-of-the-art empirical (non-certified) defenses against patch attacks achieve much lower robustness (i.e, adversarial accuracy) than our approach. In particular, local gradient smoothing (LGS) [2] achieves 24.2% empirical accuracy on CIFAR 2⨉2 while our method achieves 51.9% certified accuracy (for the largest model trained). Similarly, digital watermarking (DW) [1] achieves 15.6% empirical accuracy on CIFAR 5⨉5 while our method achieves 30.3% certified accuracy (for the largest model trained).
>
> Finally, we acknowledge that certified defenses may not be ready for real world applications.  However, our contribution makes an important step towards securing ML systems against adversarial patch attacks by substantially improving both the clean and robust accuracy achievable by a certified model.
>
>
> References
>
> [1] Jamie Hayes. On visible adversarial perturbations & digital watermarking. In Proceedings of the IEEE Conference on Computer Vision and Pattern Recognition Workshops, pp. 1597–1604, 2018.
>
> [2] Muzammal Naseer, Salman Khan, and Fatih Porikli. Local gradients smoothing: Defense against localized adversarial attacks. In 2019 IEEE Winter Conference on Applications of Computer Vision (WACV), pp. 1300–1307. IEEE, 2019.
>
> [3] Dong Su, Huan Zhang, Hongge Chen, Jinfeng Yi, Pin-Yu Chen, and Yupeng Gao. Is robustness the cost of accuracy?–a comprehensive study on the robustness of 18 deep image classification models. In Proceedings of the European Conference on Computer Vision (ECCV), pages 631–648, 2018.
>
> [4] Hongyang Zhang, Yaodong Yu, Jiantao Jiao, Eric P Xing, Laurent El Ghaoui, and Michael I Jordan. Theoretically principled trade-off between robustness and accuracy. ICML, 2019.
>
> [5] Zhang, Huan, Hongge Chen, Chaowei Xiao, Bo Li, Duane Boning, and Cho-Jui Hsieh. "Towards Stable and Efficient Training of Verifiably Robust Neural Networks." arXiv preprint arXiv:1906.06316 (2019).

---

### Author Response · Authors · 2019-11-15
**Update**

We updated the manuscript based on valuable conversations with the official reviewers:

- We include additional result supporting our main claims using experiments with a larger model

- We compare against an interesting training approach, based on a bound pooling formulation proposed by R2, which did not improve upon our top performing model.

- We report on experiments with multiple 1x1 patches, also proposed by R2, that our approach can extended to multi-patch setting. In the specific case of 1 x 1 patch, the multi-patch setting is equivalent to defending against $l_0$ attacks.

- We update all results, with only marginal differences, after fixing some minor issues in our codes.

---

### Decision · Program_Chairs · 2019-12-19

**Decision:**

Accept (Poster)

**Comment:**

This paper presents a certified defense method for adversarial patch attacks. The proposed approach provides certifiable guarantees to the attacks, and the reviewers particularly find its experiments results interesting and promising. The added new experiments during the rebuttal phase strengthened the paper. There still is a remaining concern that is novelty is limited as this paper could be viewed as the application of the original IBP to patch attacks, but the reviewers believe in that its empirical results are important.